# Energy Homeostasis-Associated (Enho) mRNA Expression and Energy Homeostasis in the Acute Stress Versus Chronic Unpredictable Mild Stress Rat Models

**DOI:** 10.3390/biomedicines11020440

**Published:** 2023-02-02

**Authors:** Mahmoud M. A. Abulmeaty, Ali M. Almajwal, Suhail Razak, Fatimah R. Al-Ramadhan, Reham M. Wahid

**Affiliations:** 1Department of Community Health Sciences, College of Applied Medical Sciences, King Saud University, Riyadh 11362, Saudi Arabia; 2Department of Medical Physiology, School of Medicine, Zagazig University, Zagazig 44519, Egypt; 3Department of Human Nutrition, College of Agriculture and Food Sciences, King Saud University, Riyadh 12372, Saudi Arabia

**Keywords:** Enho gene, chronic unpredictable stress (CUMS), acute stress, energy homeostasis

## Abstract

The energy homeostasis-associated (Enho) gene, the transcript for the Adropin peptide, is usually linked to energy homeostasis, adiposity, glycemia, and insulin resistance. Studies on Enho expression in stressful conditions are lacking. This work aimed to investigate Enho mRNA expression and energy homeostasis in acute stress (AS) versus chronic unpredictable mild stress (CUMS) rat models. A total of thirty male Wistar rats (180–220 g) were fed a balanced diet with free access to water. Rats were divided into three equal groups (*n* = 10): (a) the normal control (NC) group; (b) the AS group, where one episode of stress for 2 h was applied; and (c) the CUMS group, in which rats were exposed to a variable program of mild stressors for 4 weeks. Energy homeostasis was analyzed by the PhenoMaster system for the automatic measuring of food intake (FI), respiratory O_2_ volume (VO_2_), CO_2_ volume (VCO_2_), respiratory quotient (RQ), and total energy expenditure (TEE). Finally, liver, whole brain, and adipose (WAT) tissue samples were collected, total RNA was prepared, and RT-PCR analysis of the Enho gene was performed. The CUMS group showed higher VO_2_ consumption and VCO_2_ production, and a higher RQ than the AS group. Furthermore, the TEE and FI were higher in the CUMS group compared to the AS group. Enho gene expression in the liver, brain, and WAT was significantly higher in the CUMS group than in the AS and NC groups. We can conclude that in the chew-fed AS rats, hypophagia was evident, with a shift in the RQ toward fat utilization, with no changes in body weight despite the increase in Enho mRNA expression in all studied tissues. In the CUMS group, the marked rise in Enho mRNA expression may have contributed to weight loss despite increased FI and TEE.

## 1. Introduction

Stress is defined as a state of threatened homeostasis [1] that is associated with different physiological, neurological, and behavioral effects; this state is always accompanied by sympathetic system stimulation with increased catecholamines and glucocorticoid secretion induced by activation of the hypothalamic–pituitary–adrenal (HPA) axis, which is considered a crucial contributor in the stress response mechanism [2].

The body’s response to a particular stressor is determined by the severity and nature of the stressor, duration of exposure to the stressor, and familiarity or perception of the individual to this stressor [3]. Modern life stressors are typically chronic, including financial issues, workloads, family responsibilities, or health problems [4]. All these stressors can inversely affect our body’s homeostasis mechanisms, and their subsequent impacts can be so harmful that they predispose us to obesity and cardiac and metabolic diseases [5]. The chronic unpredictable mild stress (CUMS) rat model involves exposing rats to unpredictable episodes of mild-intensity social and physical stressors, which produce various behavioral, endocrine, and neural changes similar to major depressive disorder in humans [6].

Homeostatic mechanisms in stress usually affect the metabolism. Much attention was given to the adipose tissue peptides such as adipokines that act through both paracrine and endocrine functions. However, liver-secreted factors such as adropin were recently investigated as an energy and metabolism regulator [7]. Adropin is encoded by the energy homeostasis-associated (Enho) gene and its action is mediated by interaction with the G protein-coupled receptors (GPR19). The Enho gene is composed of one intron and two exons and is located in chromosome 9 [8]. Moreover, it is expressed in different tissues, such as the brain, heart, liver, pancreas, and coronary artery, and its high expression is observed mainly in the brain [9]

Regarding the biological regulators of adropin, many studies revealed that it is regulated by the energy status and dietary content. Many studies also found that adropin enhanced glucose oxidation and improved obesity-induced insulin resistance; in addition, adropin was found to improve endothelial cell function by increasing the expression of nitric oxide synthase, which promotes neovascularization and cardiovascular function [10]. Furthermore, another study [7] showed that transgenic overexpression or systemic adropin treatment could improve obesity, insulin resistance, and glucose tolerance. In addition, adropin attenuated components of metabolic distress associated with obesity. Adropin was reported as a hormone regulating glucose and lipid metabolism, and many different roles were investigated in hypertension, polycystic ovary syndrome, liver diseases, and cancer biology.

The advanced metabolic monitoring systems enable us to objectively measure food intake and energy expenditure in the form of total energy expenditure (TEE) based on accurate measurements of inspired oxygen volume (VO_2_) and expired carbon dioxide volume (VCO_2_). The respiratory quotient (RQ), which equals VCO_2_/VO_2_, adds more data about the utilized substrate during the test duration [11]. RQ was investigated in acute and chronic stress-subjected animal models and it was stated that substrate utilization showed distinctive forms in mice exposed to acute stress, repeated stress, and CUMS [12].

The Enho gene and its product, adropin, can be considered as a potential mediator of the metabolic system. However, its expression in conditions associated with acute or chronic stress is underinvestigated. Thus, our study aims to determine the effect of acute versus chronic stress on the expression of the Enho gene in different tissues and energy homeostasis in rat models of acute and chronic unpredictable mild stress, highlighting the role of adropin as the main contributor of energy homeostasis during these stressful conditions.

## 2. Materials and Methods

### 2.1. Animals and Procedures

In total, thirty-two male Wistar rats (185–230 g) were obtained from the animal house of the College of Applied Medical Sciences, King Saud University. Rats were randomly enrolled into two groups (*n* = 16 rats/group). The first group was (a) the acute normal control group (A-NC, *n* = 8); rats were kept under a normal light/dark cycle (6:00 a.m. to 6:00 p.m.) and 25 °C room temperature with free access to a standard rodent diet (carbohydrate 55%, fat 30%, and protein 15%) and free access to water. The second group was (b) the acute stress (AS, *n* = 8) group, where rats were exposed to one episode of stress in the form of restraining for one hour [13]. The second group was divided into (c) the chronic normal control group (C-NC, *n* = 8), where rats were kept in normal conditions without any stresses for 4 weeks, and (d) the chronic unpredictable mild stress (CUMS, *n* = 8) group, in which rats were exposed to a variable program of mild stressors for 4 weeks (Figure 1). These variable mild stressors included social defeat, cage tilting, shaking, restraining, exposure to hot air and continuous light, tail pinching, wet bedding, and food and water deprivation. The schedule and durations of these stressors are shown in Table 1 [14,15].

The study protocol was revised and approved by the Institutional Animal Care and Use Committee at King Saud University under reference no. KSU-SE-22-24 and the date 24 March 2022, and by Zagazig University under the reference number ZU-IACUC/3/F/212/2021 and the date 29 December 2021.

### 2.2. Anthropometric Measures

After acclimatization for one week, body weight was measured before and after the stress episodes. Moreover, rat length was measured and body mass index (BMI) was calculated by dividing body weight in g/nose-to-anus-length in square cm [16].

### 2.3. Indirect Calorimetry

At the end of the stress intervention period, rats were scheduled to be kept in Calo-cages of the PhenoMaster system (TSE, Berlin, Germany) individually, at normal room temperature and humidity. For acclimatization to the separate Calo-cages, the first 6 h of measurement were not used for analysis [17]. The PhenoMaster system automatically records measurements of oxygen volume (VO_2_), carbon dioxide volume (VCO_2_), respiratory quotient (RQ = VCO_2_/VO_2_), and total energy expenditure (TEE). The TEE was represented as the amount of energy expenditure in kcal/hour/kg of body weight, and kcal/hour/kg of lean body mass (LBM), which was estimated to be 0.75% of the whole-body weight [18]. Automatic food intake (FI) measurement was performed and represented as g/day [11].

### 2.4. Blood Sampling and Corticosterone Measurement

Blood samples from the tail vein were obtained at the end of the stress episodes and before indirect calorimetry measurement. Blood samples were collected in heparinized tubes (Greiner Bio-One, Germany) and then centrifuged at 500 rpm for 15 min; then, the plasma was kept frozen (−80 °C) until the time of analysis. Plasma corticosterone levels were measured using rat corticosterone ELISA kits (MyBiosource, San Diego, CA, USA; catalog number MBS761865) for confirmation of the animal model of CUMS and AS.

### 2.5. Tissue Sample Preparation

Liver, adipose (WAT) tissue, and whole-brain samples were collected in blank tubes and immediately transferred under strict freezing conditions to the central lab at Zagazig University and the lab of the Medical and Molecular Genetics Research Chair at King Saud University.

### 2.6. Gene Expression Analysis

Relative gene expression was quantified as previously reported [19]. Total RNA was isolated from the liver, adipose tissue, and brain by using an RNeasy Mini Kit (Qiagen, Hilden, Germany) and checked for purity on a NanoDrop. Approximately 1 μg of RNA template was first heat-denatured and then converted to cDNA using the RevertAid First Strand cDNA Synthesis Kit (ThermoFisher Scientific, Waltham, MA, USA). Specific primer sets for the *Enho* gene used the following primer (5′ TGCTGCTCTGGGTCATCCTCTG 3′). Fold change in gene expression was expressed, based on *actb* as a housekeeping gene, using the 2^−ΔΔCT^ method.

### 2.7. Statistical Analysis

The results were presented as means ± standard deviation (SD). Normality was tested by the Shapiro–Wilk test. The one-way ANOVA test was used to compare mean differences among study groups, in addition to a post hoc test for intergroup comparisons. The independent-sample t-test was used to compare each group with its control. SPSS, version 25 (SPSS Inc. Chicago, IL, USA), was used for analyses. Differences were considered statistically significant when the *p*-value was ≤0.05.

## 3. Results

### 3.1. Body Weight Change

As compared to the control group, the final weight and BMI were significantly decreased in the CUMS group (Table 2). After 4 weeks, rats in the CUMS group lost weight by 25.17 ± 24.46 g, while rats in the C-NC groups gained weight by approximately 7.33 ± 6.59 g. This was not the case in the AS group, considering the short duration of intervention in the AS group. Plasma corticosterone levels were significantly higher in the AS and CUMS groups versus equivalent controls, with a significant rise in the CUMS group, exceeding that in the AS group (Figure 2).

### 3.2. Indirect Calorimetry Change

Compared to equivalent controls, indirect calorimetry parameters revealed insignificant changes in the volumes of O_2_ consumption and CO_2_ production, RQ, and TEE. However, the comparison of the CUMS with the AS showed significantly higher values of VO_2_, VCO_2_, RQ, and TEE in the CUMS group (Table 3). Compared to the equivalent control, the AS group showed significant hypophagia, while the CUMS group showed significant hyperphagia. Furthermore, the FI was higher in the CUMS group compared to the AS group.

### 3.3. Enho Gene Expression

Compared to equivalent controls, the fold change of Enho gene expression was significantly increased in the AS and CUMS groups. Furthermore, the fold change of Enho gene expression was significantly higher in the CUMS than in the AS group in all studied tissues (Figure 3).

## 4. Discussion

This study demonstrated the responses to various doses of stress, taking into consideration the relationship of acute and chronic stresses with Enho gene expression in different tissues and components of energy homeostasis. One of the main contributors to the stress response process is the hypothalamus’s paraventricular nucleus (PVN), which sends stimulatory signals to the sympathetic controlling areas in the brain stem and spinal cord to enhance the release of catecholamines from both sympathetic nerve endings and adrenal medulla. In addition, PVN activates the hypothalamic–pituitary–adrenal axis (HPA) by sending stimulatory signals to the medial eminence to secrete corticotropin-releasing factor (CRF) that stimulates adrenocorticotropic hormone (ACTH) secretion from the pituitary gland, which subsequently enhances glucocorticoid release from the adrenal cortex; all these mechanisms are activated to enhance metabolism and energy homeostasis during stress [20,21].

It seems that stress can affect the energy balance in different ways: it can trigger orexigenic or anorexigenic-like responses according to the type and duration of this stressor, and it depends strongly on the response of the body to this stressor, which involves the overlapping of different hormones and factors that may synergize or counteract each other [22]. Chronic and acute stressors recruit some overlapping but also divergent systems relevant to metabolic control.

The present study showed an activated HPA axis with elevated serum corticosterone levels in the acute stress-exposed group, which may be an underlying mechanism for the decreased food intake seen in acute stress; this is consistent with the Harris study [3], which revealed that both CRF and urocortin can inhibit food intake and increase energy expenditure when infused centrally. However, we noticed decreased TEE in the AS group, which was measured 3 days after the restrain episode and indicated no further elevation of glucocorticoids after the end of the acute stress; similarly, the De Souza study showed that, in acute stress, the ACTH serum concentration reaches a peak within 5 min [23] and glucocorticoids peak within 30 min and then start to decline [24].

On the other hand, activation of the CRF by stress also results in the release of serotonin, adrenaline, and cytokines, which were shown to be good mediators in this hypophagia in acute stress [25,26]. Dallman et al. [27] stated that the balance between corticosterone and insulin is the main determinant factor of food intake and energy expenditure because the two hormones have opposing effects, and high doses of corticosteroids were shown to decrease food intake by inducing insulin release, which inhibits neuropeptide Y (NPY) expression in the hypothalamus [28]. The current study showed low TEE with no changed body weight, which can be explained by the capacity of the rats after stress to defend a new lower set-point for energy homeostasis, and so energy expenditure must also be adjusted, particularly with decreased caloric intake [22]. Another study by Dal-Zotto et al. [29] revealed that severe stress due to 20 min immobilization inhibited food intake for up to 3 days, and Harris’s study showed that single 3 h restraint increased serum corticosteroids, with high energy expenditure only during the restraint, which was corrected within an hour of the end of the restraint [30]. Furthermore, Kuti et al. [12] measured TEE during the exposure of mice to acute stress by the PhenoMaster system and reported that TEE started to increase shortly after the start of an acute stress episode. However, the current study examined TEE after the end of the acute stress episode for 3 days, providing a strong basis for the different results. Moreover, they reported that the CUMS produced a rise in TEE, which is consistent with our findings. In the current study, RQ was shifted toward fat utilization in the AS group and toward mixed substrate utilization in the CUMS group. However, the Kuti study [12] showed a shift toward fat utilization in mice exposed to acute stress and toward carbohydrate utilization under chronic various stress.

Regarding exposure to long-term stress, the impact of chronic stress on energy and metabolic homeostasis is more variable; for example, chronic stress may cause weight gain in humans, and, on the contrary, it causes weight loss in rodents exposed to psychological stress [3]. Our results showed a significant reduction in body weight in the CUMS group. This was consistent with Kuti et al. [12], who reported a reduction in body weight gain and disturbance in the body composition of mice in the form of a loss of lean mass under chronic repeated stress and loss of fat mass under chronic unpredictable stress.

Although activation of both the sympathetic nervous system and HPA system can optimize the immediate response to acute environmental stress, chronic stress induces many different mechanisms that may lead to a variety of metabolic, cardiovascular, and mental disorders [12]. There is growing evidence that chronic or repetitive stress can enhance physiological adaptation, which enables the body to restabilize in this threatening environment; this adaptation is referred to as allostasis.

Many works in the literature showed that repeated exposure to severe stressors impaired the negative feedback effect of glucocorticoids in suppressing CRF expression in the PVH, leading to the extended secretion of glucocorticoids [31], and Bhatnagar [32] stated that this loss of sensitivity to negative feedback was rostral to the PVH and pituitary gland. Moreover, Coffman et al. [33] identified the FK506 binding protein 5 (fkbp5) gene as one of the genes activated by glucocorticoid receptors, and it encodes a type of protein that inhibits the activation of these receptors by antagonizing the binding of glucocorticoids thereto, so it is considered an intracellular negative feedback mechanism that promotes adaptation by enhancing receptors’ resistance to the glucocorticoids’ action [34]. Another regulator called kruppel-like factor (klf9) was recently shown as a feedforward repressor of the glucocorticoids targeting klf2 in mouse macrophages [35]. Based on the aforementioned studies, it is found that many feedback loops work intracellularly and systemically to control and regulate HPA axis function in a chronic stress situation.

The present study showed the strong elevation of the corticosterone level in the chronic stress-exposed group compared to either the normal or acute stress-exposed one, which was not accompanied by such a significant increase in the TEE, and this can be referred to as the adaptation response and HPA axis resistance. Adaptation to high Enho gene expression during chronic stress exposure is another expected mechanism. Several reports state that adropin has a role in regulating carbohydrate and lipid substrate oxidation preferences [36]. The intraperitoneal injection of adropin reduced the high blood glucose level in a model of diet-induced obesity in mice by inhibiting hepatic glucose output and triglyceride accumulation via activation of the 5′-adenosine monophosphate-activated protein kinase (AMPK) pathway [37].

High food intake was one of the main findings in the chronic stress-exposed group, which may be attributed in part to the complex adaptive response to chronic stresses or the high expression of the Enho gene. Kuti et al. [12] reported that the suppression of food intake in mice exposed to chronic variable stresses was limited to the first 4 h after the stress episode, and then the feeding returned to the normal level during the dark period. The second assumption related to Enho gene overexpression was in line with early reports showing the relationship between adropin and high-fat-diet-fed mice [7]. Interestingly, this hyperphagia is associated with weight loss, which additionally may be explained at least in part by increased Enho gene expression. Jasaszwili et al. [38] found that adropin could suppress peroxisome proliferator-activated receptor-gamma (PPARγ), CCAAT enhancer binding protein beta (C/ebpβ), and fatty acid binding protein 4 (Fabp4) mRNA expression in primary preadipocytes in the rat, leading to a reduction in lipid accumulation. However, it was observed that rats were losing weight instead of developing obesity, and this interesting finding can be explained by Souza [39], who showed that exposure to stressors in rats for 5 weeks impaired kidney function, with a significant loss of weight as a result of reduced kidney volumes and water loss.

Adropin is considered one of the main peptides that has a great impact on metabolism and energy homeostasis; it is highly expressed in the brain, particularly in areas controlling complex behaviors such as circadian rhythms and the stress response [9]. The present study showed that the highest expression of adropin mRNA was found in the chronic stress group, and it was noticed that RQ was shifted to carbohydrate utilization in the CUMS compared to the AS group, despite insulin and glucocorticoid resistance. This can be explained by Zhang [9], who revealed that adropin promotes glucose uptake by the tissue through increasing the expression of glucose transporter (GLUT 1) receptors. In addition, it enhances glucose oxidation by improving metabolic intolerance in obese and insulin-resistant mice; this mechanism was reinforced by the suppression of fatty acid oxidation as a result of the downregulation of carnitine palmitoyltransferase-1B (CPT-1B) and cluster of differentiation 36 (CD36).

The contribution of adropin was investigated as well by Gao et al. [40], who reported that exogenous adropin causes an increase in insulin receptor substrate 1 (IRS1, IRS2) and AKT phosphorylation in obese mice, indicating improved hepatic insulin sensitivity [8,40]. Furthermore, Douglas et al. reported that stress granules during nutrient stress conditions such as long-term starvation can regulate fatty acid oxidation and lipid metabolism, as they stimulate the depletion of mitochondrial voltage-dependent anion channel (VDAC) porins, resulting in a marked decrease in fatty acids moved into mitochondria for oxidation [41].

The present study reported that high adropin expression was observed in adipose tissue, as well as during chronic stress, indicating that adropin contributes to the adipocytes’ function by suppressing the differentiation of 3T3-L1 cells into mature adipocytes [38], thus interfering with lipogenesis.

Eventually, the body’s metabolism during chronic stress is different from that in acute stress as it is regulated by many complex overlapping mechanisms, aiming to maintain the energy balance and regulate systems’ functions, and the development of adaptation is one of the main mechanisms to resist stress’ long-term consequences on bodily functions. However, long-term stress can disturb these compensatory plans and result in failure to maintain this balance due to the dominant underlying mechanisms that eventually shape different individuals’ responses during stress. Moreover, the human reaction to depression is different from the rat’s response to CUMS. It was reported that patients with coronary heart disease and depression have low serum adropin levels, as well as a negative correlation between adropin levels and the Patient Health Questionnaire-9 [42].

Despite many strengths, this study has some limitations, such as a lack of body composition analysis to identify which body compartment was lost in the CUMS group. Moreover, we did not measure plasma adropin levels and substrate levels such as glucose and lipid panels. The apparent difference in the weight changes in the A-NC and C-NC groups is due to the duration factor. Rats in the A-NC stayed for a short duration (4 days only). This duration included 3 days of individual housing in the Calo-cages of the TSE PhenoMaster system for the measurement of TEE. This social isolation produced some stress that may have caused this weight loss. The rats in C-NC remained unstressed in the usual housing conditions in groups of five rats for 4 weeks, so they displayed some weight gain. However, this study provides a sufficient answer to the research questions and offers a basis for future research.

## 5. Conclusions

In conclusion, different doses of stress in rat models produce distinctive impacts on energy homeostasis. In the AS rats, hypophagia was evident, with a shift in the RQ toward fat utilization and with no significant changes in body weight despite the increase in Enho mRNA expression in all studied tissues. In the CUMS group, the marked rise in Enho mRNA expression may have contributed to weight loss despite the increased FI and TEE. This work may support the role of the Enho gene, which encodes adropin, in obesity prevention under long-term stressful conditions in rats.

## Figures and Tables

**Figure 1 biomedicines-11-00440-f001:**
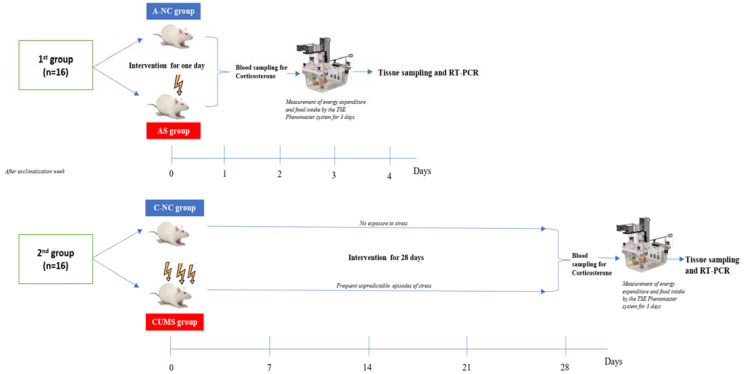
Schematic representation of the experimental design.

**Figure 2 biomedicines-11-00440-f002:**
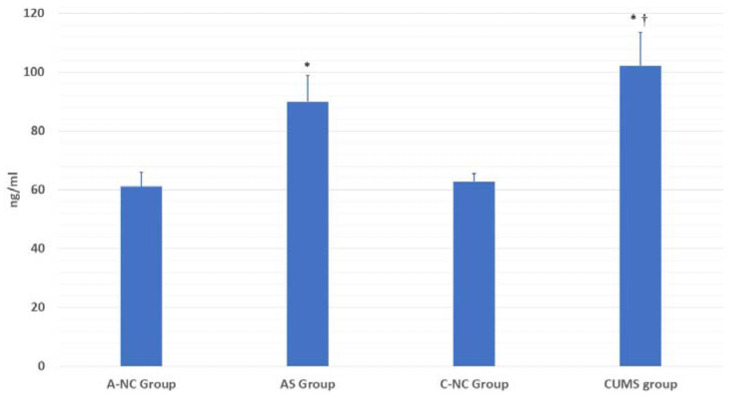
Corticosterone plasma levels after the AS and CUMS versus control. The asterisk represents statistical differences versus control (*p* < 0.05). † denotes the significant difference between CUMS and AS groups (*p* < 0.01).

**Figure 3 biomedicines-11-00440-f003:**
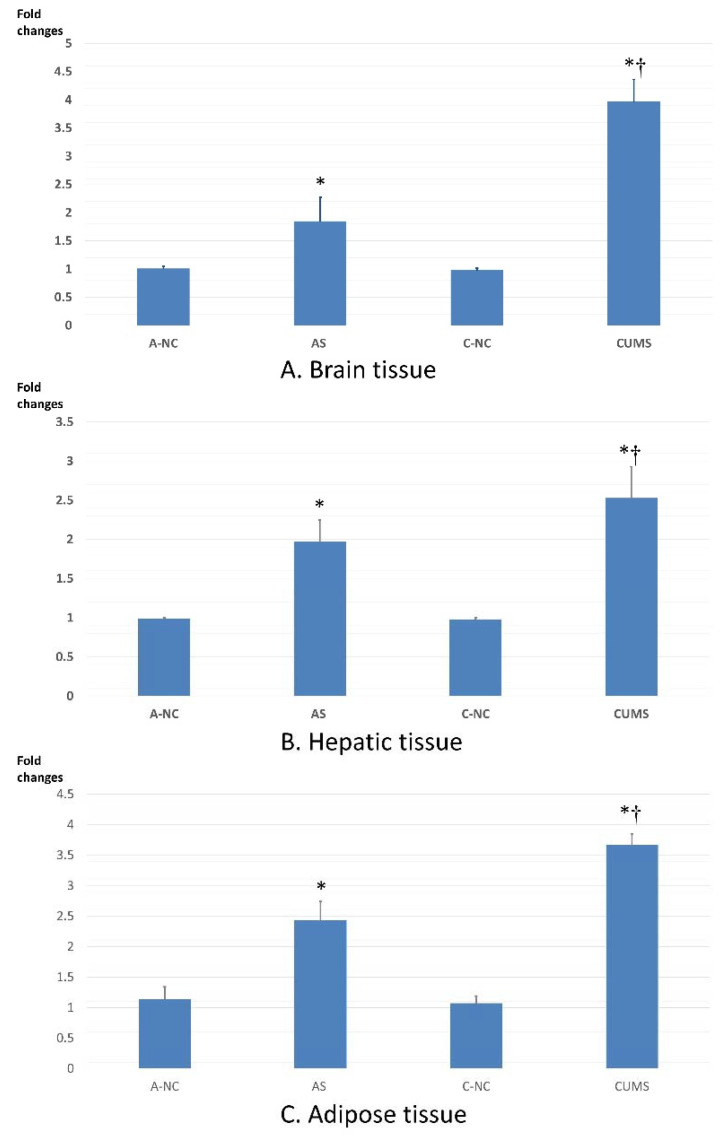
Fold changes in Enho gene expression in the brain (**A**), liver (**B**), and adipose tissue (**C**) of all studied groups. * represents significant differences versus the equivalent control, † denotes significant differences between CUMS and AS groups (*p* < 0.05).

**Table 1 biomedicines-11-00440-t001:** Schedule and durations of variable mild stressors used in the study.

Day	Week 1	Week 2	Week 3	Week 4
Sunday	Cage tilting 24 h	Social defeat 30 min	Cage tilting 24 h	Social defeat 30 min
Monday	Shaking at 150 rpm for 1 h	Restraint for 1 h	Shaking at 150 rpm for 1 h	Restraint for 1 h
Tuesday	Restraint for 1 h	Hot air for 10 min	Restraint for 1 h	Hot air for 10 min
Wednesday	Hot air for 10 min	Tail pinch 2 min	Hot air for 10 min	Tail pinch 2 min
Thursday	Continuous light 12 h	Wet bedding 24 h	Continuous light 12	Wet bedding 24 h
Friday	Food derivation 24 h	Continuous light 12	Wet bedding 24 h	Continuous light 12
Saturday	Water derivation 24 h	Food derivation 24 h	Food derivation 24 h	Water derivation 24 h

**Table 2 biomedicines-11-00440-t002:** Body weight changes among study groups.

Variables	A-NC Group Mean ± SD (*n* = 8)	AS Group Mean ± SD (*n* = 8)	*p*-Value	C-NC Group Mean ± SD (*n* = 8)	CUMS Group Mean ± SD (*n* = 8)	*p*-Value	*p*-Value
Rat length (cm)	20.85 ± 0.25	20.91 ± 0.66	0.778	20.83 ± 0.26	20.92 ± 0.66	0.778	0.983
Baseline weight (g)	211.33 ± 14.79	209.17 ± 16.34	0.868	213.50 ± 15.63	215.00 ± 35.51	0.908	0.971
Final weight (g) *	201.00 ± 14.45	194.50 ± 18.52	0.569	220.83 ± 21.75	189.83 ± 22.08	0.012	0.057
Baseline BMI (g/cm^2^)	0.49 ± 0.03	0.48 ± 0.04	0.831	0.50 ± 0.04	0.49 ± 0.08	0.831	0.925
Final BMI (g/cm^2^)	0.46 ± 0.03	0.44 ± 0.04	0.501	0.51 ± 0.05	0.45 ± 0.07	0.042	0.110
Weight gain (g)	−10.33 ± 21.03	−14.67 ± 20.01	0.871	7.33 ± 6.59	−25.17 ± 24.46	0.050	0.166

A-NC = normal control for the acute stress group, AS = acute stress group, C-CN = normal control for the CUMS group, CUMS = chronic unpredictable mild stress group, and MBI = body mass index. * Final weight was measured after 3 days from the baseline weight in the AS group and after 31 days from the baseline in the CUMS group.

**Table 3 biomedicines-11-00440-t003:** Indirect calorimetry parameters of all studied groups.

Variables	A-NC Group Mean ± SD (*n* = 8)	AS Group Mean ± SD (*n* = 8)	*p*-Value	C-NC Group Mean ± SD (*n* = 8)	CUMS Group Mean ± SD (*n* = 8)	*p*-Value	*p*-Value ^†^
VO_2_ (mL/h/kg)	2365.84 ± 406.16	2032.67 ± 205.48	0.071	2369.13 ± 388.11	2369.95 ± 93.75	0.996	0.172
VO_2_ (mL/h/kg LBM)	1561.63 ± 272.83	1333.36 ± 141.96	0.069	1553.27 ± 264.28	1599.14 ± 67.71 *	0.703	0.137
VO_2_ (mL/h/rat)	450.60 ± 91.52	376.68 ± 47.58	0.090	451.44 ± 88.05	493.22 ± 47.99 *	0.326	0.071
VCO_2_ (mL/h/kg)	1810.26 ± 427.08	1489.90 ± 240.49	0.105	1802.95 ± 416.54	1886.21 ± 114.44 *	0.664	0.192
VCO_2_ (mL/h/kg LBM)	1195.56 ± 286.16	977.72 ± 161.83	0.101	1191.53 ± 279.58	1272.86 ± 80.81 *	0.528	0.146
VCO_2_ (mL/h/rat)	345.53 ± 91.33	276.54 ± 49.81	0.112	345.78 ± 89.24	392.71 ± 43.61 *	0.272	0.076
RQ	0.76 ± 0.07	0.73 ± 0.05	0.366	0.76 ± 0.05	0.79 ± 0.02 *	0.225	0.222
TEE (kcal/h/kg)	11.33 ± 2.07	9.66 ± 1.08	0.087	11.29 ± 2.01	11.43 ± 0.50	0.881	0.184
TEE (kcal/h/kg LBM)	7.48 ± 1.39	6.34 ± 0.74	0.077	7.49 ± 1.36	7.71 ± 0.36 *	0.718	0.138
TEE (kcal/h/rat)	2.16 ± 0.46	1.79 ± 0.24	0.092	2.13 ± 0.43	2.38 ± 0.24 *	0.252	0.071
Food intake (g/rat)	22.95 ± 4.37	16.60 ± 2.69	0.027	20.65 ± 5.47	28.69 ± 5.34 *	0.007	0.004

VO_2_ = volume of consumed oxygen, VCO_2_ = volume of produced carbon dioxide, LBM = lean body mass, RQ = respiratory quotient (VCO_2_/VO_2_), and TEE = total energy expenditure. ^†^ The *p*-value of the ANOVA test, * significant in comparison to the AS group by LSD post hoc test.

## Data Availability

The raw data supporting the conclusions of this article will be made available by the authors, without undue reservation, to any qualified researcher.

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
