# Peer review of "Energy Homeostasis-Associated (Enho) mRNA Expression and Energy Homeostasis in the Acute Stress Versus Chronic Unpredictable Mild Stress Rat Models"

_biomedicines, 2023, doi:10.3390/biomedicines11020440_

Round 1
Reviewer 1 Report
The biomedicines-2145240 manuscript is interesting and aims to clarify the levels and relationship with energy homeostasis of Energy homeostasis-associated (Enho) mRNA expression in the acute versus chronic stress.
Some issues need to be clarified before considering publication of the manuscript.
All evaluations were done after acute stress ((A-NC group n=8) and AS group (n=8)) or after 4 weeks ((C-CN n=8) and CUMS group n=8)) . Did the authors verify the statistical power of the study?
Table 2 shows that the first group lost weight (but also the control). To justify.
Did the CUMS group lose weight (loss of fat mass due to psychophysical stress?). To justify.
From the analysis of the metabolimeter (Table 3) it is deduced that the small difference in the weight loss of the AS group compared to the A-NC may be due to a decrease in feeding. To justify. Is it due to physical stress?
From the same table it can be seen that the CUMS group, on the other hand, fed more. To support the loads induced by stressors? To justify.
From Figure 3 it appears that acute stress and chronic stress induce the same level of corticosterone release. It's curious.
Adropin expression levels in the brain, liver and adipose tissue increase with stresses. In chronic stress more than in acute stress. However, putting only this parameter in relation to the effects found seems somewhat limiting in my opinion.
Author Response
Response to Reviewers
Manuscript ID: biomedicines-2145240
Manuscript Title: “Energy homeostasis-associated (Enho) mRNA expression and energy homeostasis in the acute stress versus chronic unpredictable mild stress rat models”
We thank the reviewers for their careful examination of the manuscript and appreciate the useful suggestions to improve the quality of our paper. Our point-by-point response to the reviewers' comments is given below. Changes in the manuscript are marked in red color. Please note that the pages and line numbers mentioned in the reviewers’ comments refer to the original manuscript, whereas those in the authors’ reply refer to the revised manuscript. By the end, the manuscript word count exceeded 4000 words as was recommended by the editor.
Comments from the Reviewers:
Reviewer 1
- The biomedicines-2145240 manuscript is interesting and aims to clarify the levels and relationship with energy homeostasis of Energy homeostasis-associated (Enho) mRNA expression in acute versus chronic stress.
Response: Perfect summarization, thanks.
----------------------------------------------------
- Some issues need to be clarified before considering the publication of the manuscript.
All evaluations were done after acute stress ((A-NC group n=8) and AS group (n=8)) or after 4 weeks ((C-CN n=8) and CUMS group n=8)). Did the authors verify the statistical power of the study?
Response: Thanks for this note. The difference between acute and chronic stress is the duration, i.e. the AS group was exposed to one stressful episode in one day while the CUMS group was exposed to variable stressors for 28 days. To find out the effects properly, we created an equivalent control group for each stress dose i.e. A-NC for AS and C-NC for CUMS. The investigations for A-NC and AS were done after acute stress (i.e. 0-4 days after the stress episode), while that for C-NC and CUMS were taken after the end of 4 weeks of stress exposure. The time mapping was added to figure 1.
The sample size and statistical power were calculated based on the acceptable range of the DF, (Arifin and Zahiruddin, 2017). The DF was replaced with the minimum (10) and maximum (20) DFs to obtain the minimum and maximum numbers of rats per group based on an independent sample t-test where k=2:
Minimum n=10/k+1 Maximum n=20/k+1
So, the minimum and maximum numbers of animals required are:
Minimum n=10/2+1= 6 rats Maximum n=20/2+1 = 11 rats
Accordingly we used 8 rats to be suitable with the number of Calo cages of the Phenomaster system in our lab.
Arifin WN, Zahiruddin WM. Sample Size Calculation in Animal Studies Using Resource Equation Approach. Malays J Med Sci. 2017 Oct;24(5):101-105. doi: 10.21315/mjms2017.24.5.11. Epub 2017 Oct 26. PMID: 29386977; PMCID: PMC5772820.
----------------------------------------------------
- Table 2 shows that the first group lost weight (but also the control).To justify.
Response: Thanks for this comment. The A-NC and AS groups lost 10.33±21.03 and -14.67±20.01 respectively however this change was insignificant. So I think many factors can affect their weights on the first days like habituation, water intake, water output, and physical activity.
----------------------------------------------------
- Did the CUMS group lose weight (loss of fat mass due to psychophysical stress?).To justify.
Response: Thanks for this comment. Unfortunately, we did not do a body composition analysis to know which compartment was lost. This was added as a limitation to this study. The focus of this work was slightly different. However, as mentioned in the discussion section, lines 286-290, it was explained by loss of water mainly because in this case we have nearly constant TEE and Souza study showed that submission of stressors to rats for 5 weeks impaired kidney function with significant loss of weight as a result of reduced kidney volumes and water loss.
----------------------------------------------------
- From the analysis of the metabolimeter (Table 3) it is deduced that the small difference in the weight loss of the AS group compared to the A-NC may be due to a decrease in feeding.To justify. Is it due to physical stress?
Response: thanks for this note. Yes, the significant reduction of food intake in the AS group versus the A-NC groups is the main cause of the noticed nonsignificant weight loss. However, changes in the TEE when comparing the A-NC and AS groups were insignificant (P-value was 0.077), which means that stress may need more time to change weight in a significant value.
----------------------------------------------------
- From the same table it can be seen that the CUMS group, on the other hand, fed more.To support the loads induced by stressors? To justify.
Response: Sure, from frequent monitoring of feeding during the intervention period in the CUMS group, the food intake was reduced early in the stress period and then increased starting from the third week (unpresented data). Besides, we only measured food intake automatically by the Phenomaster system at the end of the experiments. The reported hyperphagia and insignificant change of TEE in the CUMS group may be explained by the rise of Enho gene expression or may be due to the development of stress-induced insulin resistance that increased NPY levels in the hypothalamus resulting in an increase in food intake. The discussion was updated for more clarification on this point.
----------------------------------------------------
- From Figure 3 it appears that acute stress and chronic stress induce the same level of corticosterone release.It's curious.
Response: Sorry for this oversight. The corticosterone level in the CUMS group was much higher than its level in the acute stress group. Means±SD in theCUMS vs AS were 102.27±2.55 vs 89.97±3.16, respectively. The t-value was -6.76823. The p-value was 0.000071. figure 2 was updated to clearly show that. This was discussed in lines 265-267. The main difference in both doses of stress is the time factor. In the CUMS, the occurrence of glucocorticoid resistance or adaptation response in the HPA axis may occur.
----------------------------------------------------
- Adropin expression levels in the brain, liver, and adipose tissue increase with stress.In chronic stress more than in acute stress. However, putting only this parameter in relation to the effects found seems somewhat limiting in my opinion.
Response: The metabolism during chronic stress is regulated by many complex overlapping mechanisms aiming to keep energy balance and regulate systems’ functions. This includes the contribution of different hormones and adipokines. However, adropin was shown to have many compensatory mechanisms for this stress condition that makes it a potential target for alleviating chronic stress response consequences. The discussion was updated to clarify these relationships.
----------------------------------------------------
Author Response
Response to Reviewers
Manuscript ID: biomedicines-2145240
Manuscript Title: “Energy homeostasis-associated (Enho) mRNA expression and energy homeostasis in the acute stress versus chronic unpredictable mild stress rat models”
We thank the reviewers for their careful examination of the manuscript and appreciate the useful suggestions to improve the quality of our paper. Our point-by-point response to the reviewers' comments is given below. Changes in the manuscript are marked in red color. Please note that the pages and line numbers mentioned in the reviewers’ comments refer to the original manuscript, whereas those in the authors’ reply refer to the revised manuscript.
Comments from the Reviewers:
Reviewer 2:
- To make readers easy to read and understand, please provide abbreviations in this article as first time they appear in each of three sections: the abstract; the main text; the first figure or table.
Response: Sorry for this oversight. We confirm that all abbreviations were mentioned in full detail at the first mention.
----------------------------------------------------
- Please correct the spelling for statistical in figure 2, line number 150.
Response: Sorry for this oversight. It is corrected.
----------------------------------------------------
- Please correct the subscript or superscript font for VCO2 and VO2 in line number 161 and 162 (Table 3). Also, authors are suggested to write the consistent p-value in all articles suggested by journal.
Response: Sorry for this oversight. It is corrected, thanks.
----------------------------------------------------
- In Figure 3, line number 170 to 172, please clearly mention the meaning of * and † with symbol, which represents the specified figure.
Response: Done, thanks.
----------------------------------------------------
- Authors are suggested to add some information about VCO2 and VO2 in the context of acute and chronic stress in the introduction section.
Response: Done at lines 71 to 77, thanks
----------------------------------------------------

Round 2
Reviewer 1 Report
The authors responded to most of the critical issues encountered.
However, one substantial point still remains to be clarified.
After the period of stress, individuals in the CUMS group lose weight while the C-NCs do not, indeed appear to gain.
This control should actually be somewhat similar to A-NC.
In fact, this C-NC group is rather strange if referred to the A-NC. Or is it the A-NC group that is strangely losing weight instead?
Cortisol for these two control groups is equal.
The same goes for the expression of the Enho gene.
If these parameters do not change in these two control groups, why these differences in diet and weight?
This discrepancy is quite constraining for discussion of the data.
Author Response
Response to Reviewers _ 2nd round
Manuscript ID: biomedicines-2145240
Manuscript Title: “Energy homeostasis-associated (Enho) mRNA expression and energy homeostasis in the acute stress versus chronic unpredictable mild stress rat models”
We thank the reviewers for their careful examination of the manuscript and appreciate the useful suggestions to improve the quality of our paper. Our point-by-point response to the reviewers' comments is given below. Changes in the manuscript are marked in blue color. Please note that the pages and line numbers mentioned in the reviewers’ comments refer to the original manuscript, whereas those in the authors’ reply refer to the revised manuscript.
Comments from the Reviewers:
Reviewer 1
- The authors responded to most of the critical issues encountered.
Response: Thanks for your guidance.
----------------------------------------------------
- However, one substantial point still remains to be clarified.
After the period of stress, individuals in the CUMS group lose weight while the C-NCs do not, indeed appear to gain.
This control should actually be somewhat similar to A-NC.
In fact, this C-NC group is rather strange if referred to the A-NC. Or is it the A-NC group that is strangely losing weight instead?
Cortisol for these two control groups is equal.
The same goes for the expression of the Enho gene.
If these parameters do not change in these two control groups, why do these differences in diet and weight?
This discrepancy is quite constraining for discussion of the data.
Response: Thanks for this comment. Regarding body weight, rats in the A-NC stayed for a short duration (4 days only). This duration included 3 days of individual housing in the Calo cages of the TSE Phenomaster system for measurement of TEE. This social isolation produced some stress that may produce this weight loss. The rats in C-NC stayed unstressed in usual conditions for 4 weeks so they catch some weight gain. The rise in corticosterone levels is due to the stimulation of the HPA axis by stress. Moreover, both A-NC and C-NC did not expose to any strong stresses so it is expected to have similar corticosterone levels. Similarly, both have close values of the Enho gene expression. This was added to the discussion and marked in blue at lines 328-333.
----------------------------------------------------
Round 3
Reviewer 1 Report
The authors have responded sufficiently to the critical issues encountered.
The manuscript can be accepted for publication.